# The Effect of Single and Combined Use of Gamma Radiation and Ethylmethane Sulfonate on Early Growth Parameters in Sorghum

**DOI:** 10.3390/plants9070827

**Published:** 2020-06-30

**Authors:** Maliata Athon Wanga, Hussein Shimelis, Lydia N. Horn, Fatma Sarsu

**Affiliations:** 1School of Agriculture, Earth and Environmental Sciences, University of KwaZulu-Natal, Private Bag X01, Scottsville 3209, Pietermaritzburg, South Africa; shimelish@ukzn.ac.za; 2Directorate of Agricultural Research and Development, Ministry of Agriculture, Water and Land Reform, Private Bag 13184, Windhoek, Namibia; 3Multidisciplinary Research Centre, University of Namibia, Private Bag 13301, Windhoek, Namibia; lnhorn@yahoo.com; 4Plant Breeding and Genetics Section, Department of Nuclear Sciences and Applications, International Atomic Energy Agency (IAEA), Vienna International Centre, P.O. Box 100, 1400 Vienna, Austria; F.Sarsu@iaea.org

**Keywords:** ethylmethane sulfonate, gamma radiation, sorghum

## Abstract

Success in inducing genetic variation through mutagenic agents is dependent on the source and dose of application. The objective of this study was to determine the optimum doses of a single and combined use of gamma radiation and ethylmethane sulfonate (EMS) for effective mutation breeding in sorghum. The study involved two concurrent experiments as follows: in experiment I, the seeds of four sorghum genotypes (‘Parbhani Moti’, ‘Parbhani Shakti’, ‘ICSV 15013′, and ‘Macia’) were treated using gamma radiation (0, 300, 400, 500 and 600 Gy), EMS (0, 0.5 and 1.0%), and gamma radiation followed by EMS (0 and 300 Gy and 0.1% EMS; 400 Gy and 0.05% EMS). In experiment II, the seeds of two genotypes (‘Macia’ and ‘Red sorghum’) were treated with seven doses of gamma radiation only (0, 100, 200, 300, 400, 500 and 600 Gy). Overall, the combined applied doses of gamma radiation and EMS are not recommended due to poor seedling emergence and seedling survival rate below LD_50_. The best dosage of gamma radiation for genotypes Red sorghum, Parbhani Moti, Macia, ICSV 15013 and Parbhani Shakti ranged between 392 and 419 Gy, 311 and 354 Gy, 256 and 355 Gy, 273 and 304 Gy, and 266 and 297 Gy, respectively. The EMS optimum dosage ranges for genotypes Parbhani Shakti, ICSV 15013, Parbhani Moti and Macia were between 0.41% and 0.60%, 0.48% and 0.58%, 0.46% and 0.51%, and 0.36% and 0.45%, respectively. The above dose rates are useful to induce genetic variation in the tested sorghum genotypes for greater mutation events in sorghum breeding programs.

## 1. Introduction

Use of climate-smart crop cultivars is a key mitigation strategy against the unpredictable impacts of climate change associated with threats such as drought and heat stress, flooding, soil erosion, and salinization [1]. Sorghum (*Sorghum bicolor* [L.] Moench) is one of the key crops adapted to grow under arid and semiarid and harsh conditions where other crops fail to survive under similar environmental conditions. This makes sorghum a crop of choice in sub-Saharan Africa (SSA), Asia and similar agro-ecologies. However, yields in sorghum are low in SSA, with a mean grain yield level of less than 1 t/ha far below the potential grain yield of the crop, which reaches up to more than 4 t/ha [2]. Therefore, plant breeders need to broaden the genetic variation of sorghum to develop high-performing cultivars with enhanced grain and biomass yields in SSA.

Natural genetic variation is created via spontaneous mutation. However, the frequency of natural mutation is very low, varying from 10^−5^ to 10^−8^ per loci in higher plants [3]. Induced genetic variation using physical and chemical mutagens is an alternative method of creating the new genetic variation needed for the breeding of climate-resilient crop cultivars. Different mutagenic agents have a variable effect on mutations. For example, physical mutagens such as gamma irradiation result in deletion, translocation and aberrations of chromosomes [4]. Meanwhile, chemical mutagens such as ethylmethane sulfonate (EMS) results in modification of certain nucleotides [5,6]. The use of artificial mutagenesis emulates the natural process of spontaneous mutation by maintaining the genetic status quo and without introducing alien genes, unlike in genetically modified organisms [7,8,9]. Induced mutation increases the proportion of genetic variation multi-fold (1000 to a million times) [10,11,12]. Artificial mutagenesis coupled with innovative breeding approaches such as speed breeding techniques and use of single seed descent selection method can deliver desired cultivars with a relatively shorter breeding cycle [13,14,15]. 

Success in inducing genetic variation using mutagenic agents is dependent on the source and dose of application [16,17,18]. Gamma radiation and EMS are widely used mutagenic agents to create genetic variation in plant breeding programs [19,20,21]. The use of gamma radiation and EMS in sorghum breeding programs has been reported by several researchers. For example, in Nigeria, eight sorghum mutant lines were developed by irradiating seed at a dose rate of 300 Gy [22]. In Mali, genetic variation was achieved for agronomic traits including plant height, drought tolerance, grain quality and maturity period when sorghum seeds were irradiated with a range of 200 and 300 Gy [23]. In Japan, sorghum seed irradiated at 400 Gy produced brown midrib (bmr) lines, a phenotype associated with low lignin content and increased digestibility [24]. In the USA, nuclear male sterility, a trait important for hybridization, was generated when seeds were treated with EMS doses ranging between 0.1 and 0.3% [25]. Bloomless (bm) mutant lines, linked with resistance to the greenbug pest and sheath blight disease in sorghum, were produced when the seed was treated with EMS doses ranging between 0.1 and 0.6% [26,27]. From the foregoing, it can be concluded that variable ranges of gamma radiation and EMS doses were used in different crop genotypes. Hence, there is a need to select optimum doses for each mutagenic source for specific sorghum genotypes to achieve a higher proportion of mutation events and genetic variation. 

The mutant variety database (MVD) holds records of 3320 varieties officially released across 73 countries and 228 plant species by different nations of the world [28]. Single treatments of plant materials with gamma radiation and EMS are the most studied mutagenic agents. Gamma radiation is the most preferred mutagenic agent because of its relatively higher degree of plant tissue penetration, reproducibility and greater mutation frequency [9,28]. It was reported that gamma radiation accounted for 1666 of mutant plant varieties on the MVD [28]. In Indonesia, Human et al. [29] developed three genetically stable high-yielding varieties through gamma irradiation. EMS is the most preferred chemical mutagenic agent and requires relatively simple equipment and facilities [19,20]. EMS was used to generate 107 of the mutant plant varieties in the MVD. This demonstrates the significance of mutation breeding as an important tool to enhance the genetic diversity of plant species. 

The combined use of physical and chemical mutagens has been used to develop 37 of the mutant plant varieties in the MVD. However, there is a limited record of sorghum mutant varieties developed by combined treatment of gamma radiation and EMS in the MVD. Reports reveal that combined use of gamma radiation and EMS might yield higher mutation frequency than using a single mutagenic agent [30,31]. Thus, the optimum dose of single or combined use of gamma radiation and EMS needs to be assessed and established for large-scale mutagenesis [17,32]. Higher doses of gamma radiation and EMS lead to undesirably low seed germination; poor seedling emergence, survival, growth and development; and reduced flowering, seed set, and seed viability [16,28,33]. A dose of a mutagenic treatment which results in a mean lethal dose of 50% (LD_50_) and a mean growth reduction of 50% (GR_50_) is suggested to provide a higher chance of producing effective mutations and mutagenic events for targeted selection [9,34]. Therefore, the objective of this study was to determine the effect of a single or combined use of gamma radiation and ethylmethane sulfonate (EMS) for effective mutagenesis and breeding in sorghum.

## 2. Results

### 2.1. Experiment I

#### 2.1.1. Effect of Single Doses of Gamma Radiation 

Highly significant interaction effects (*p* < 0.01) were recorded between genotype and gamma radiation dosage on seedling emergence percentage and survival percentage (Table 1). This suggests that the optimum dose of gamma radiation in sorghum mutation breeding is significantly influenced by the genotype. Table 2 summarizes the mean of seedling emergence percentage, survival rate and shoot length. Results showed a significant reduction in assessed traits with increased doses of gamma radiation when compared with their respective controls (Table 2). Genotype Macia showed the lowest seedling emergence value of 2.4% at 600 Gy, showing non-significant differences when compared with genotypes ICSV 15013, Parbhani Shakti and Parbhani Moti with values of 5.2%, 4.6% and 4.4%, respectively. Genotype Macia could not survive doses ≥ 500 Gy, whereas genotypes Parbhani Moti, ICSV 15013 and Parbhani Shakti had relatively better survival rates at 600 Gy, 5.8%, 5.3% and 2.2%, respectively. The shoot length was significantly decreased by doses of ≥400 Gy in all genotypes when compared to controls (Table 2). The optimum radiation doses aiming at LD_50_ for seedling emergence for genotypes Parbhani Moti, ICSV 15013, Parbhani Shakti and Macia were 311, 304, 297, and 278 Gy, respectively (Figure 1). The LD_50_ for seedling survival rate for genotypes Parbhani Moti, ICSV15013, Parbhani Shakti and Macia were 354, 273, 266 and 256 Gy, respectively. The optimum radiation dose aiming for GR_50_ for shoot length for genotypes ICSV15013, Parbhani Shakti, Macia and Parbhani Moti were 521, 510, 479 and 436 Gy, respectively (Figure 1). 

#### 2.1.2. Effect of Single Doses of EMS

Significant interactions (*p* < 0.01) were observed between genotype and EMS concentrations on seedling emergence percentage and survival rate (Table 1). This suggests that the optimum EMS dose in sorghum mutation breeding is significantly influenced by the genotype. The means of seedling emergence percentage, survival percentage and shoot length showed a significant reduction with increased EMS concentration (Table 2). The lowest seedling emergence value, 4.6%, was recorded in genotype Parbhani Moti at 1% EMS, showing significant differences when compared with the 17.6% and 11.0% recorded in genotypes Parbhani Shakti and ICSV 15013, respectively. The lowest seedling survival rate value, 2.4%, was recorded in genotype Macia at 1% EMS, showing non-significant differences when compared with genotypes Parbhani Shakti, ICSV 15013 and Parbhani Moti with values of 6.4%, 4.8% and 3.0%, respectively. The LD_50_s for seedling emergence for genotypes Parbhani Shakti, ICSV15013, Parbhani Moti and Macia were 0.60%, 0.58%, 0.46% and 0.45%, respectively (Figure 2). The LD_50_s for seedling survival rate for genotypes Parbhani Moti, ICSV 15013, Parbhani Shakti, and Macia were 0.51%, 0.48%, 0.41% and 0.36% EMS, respectively. The GR_50_s for shoot length for genotype ICSV 15013, Parbhani Shakti, Parbhani Moti, and Macia were 1.35%, 0.90%, 0.87% and 0.67% EMS, respectively (Figure 2).

#### 2.1.3. Effect of Combined Doses of Gamma Radiation and EMS

Highly significant interactions (*p* < 0.01) were observed between genotype and combined use of gamma radiation doses and EMS concentrations on seedling emergence percentage and survival percentage (Table 1). The mean of seedling emergence percentage, survival percentage and shoot length showed a significant reduction with increased combined dosage (Table 2). Seedling emergence percentages of genotypes Parbhani Moti, Macia, Parbhani Shakti and ICSV 15013 at dose rates of 300 Gy and 0.1% EMS were 27.6%, 19.7%, 16.8% and 14.3%, respectively (Figure 3). The lowest seedling emergence value, 1%, was recorded in genotype Macia at dose rates of 400 Gy and 0.05% EMS (Table 2). Genotype Macia could not survive a dose rate of 400 Gy and 0.05% EMS, compared with Parbhani Moti, ICSV 15013 and Parbhani Shakti, which survived with 1.4%, 1.0% and 0.6%, respectively. Genotype Parbhani Moti showed the lowest mean shoot length value of 10.3 cm, recorded at dose rates of 400 Gy and 0.05% EMS, showing non-significant differences when compared with genotypes ICSV 15013 and Parbhani Shakti, with 12.8 and 12.4 cm, respectively. The growth reduction of shoot length for genotypes Parbhani Shakti, ICSV 15013, Parbhani Moti and Macia at the dose rate of 300 Gy and 0.1% EMS were 64.6%, 57.3%, 53.1% and 44.9% of control, respectively (Figure 3).

### 2.2. Experiment II

Significant interactions (*p* < 0.01) were observed between genotype and gamma radiation dosage on germination percentage, seedling emergence percentage, seedling survival percentage, number of panicles per m^2^, number of productive panicles per m^2^, panicle length, plant height and seed viability (Table 3). Table 4 summarizes the means of traits studied. The germination percentage for genotype Macia was significantly reduced by doses ≥300 Gy while there was a non-significant reduction in the genotype Red sorghum (Table 4). The lowest seedling emergence was recorded in genotype Macia (0.7%) at a dose of 600 Gy showing significant difference compared to genotype Red sorghum (13.9%). The seedling survival rate and shoot length demonstrated a similar trend as emergence percentage. Genotype Macia could not survive at 600 Gy as compared with Red sorghum, which survived the dose with 9.2% (Table 4). The mean of shoot length was significantly decreased by doses of ≥400 Gy in both genotypes. The LD_50_s for germination percentage for genotypes Red sorghum and Macia were 6158 and 1092 Gy, respectively (Figure 4). The LD_50_s for seedling emergence for the genotypes Red sorghum and Macia were 419 and 355 Gy, respectively (Figure 4). The LD_50_s for seedling survival rate for genotypes Red sorghum and Macia were 392 and 288 Gy, respectively. The GR_50_s for shoot length for genotypes Red sorghum and Macia were 686 and 658 Gy, respectively (Figure 4).

The number of panicles m^−2^ was significantly reduced by doses ≥400 and 600 Gy in Macia and Red sorghum, respectively (Table 4). The number of panicles m^−2^ in Red sorghum was significantly increased at 100 Gy and decreased at 600 Gy, revealing different responses of genotypes to gamma radiation doses. Plant height of the genotype Red sorghum was significantly reduced by doses ≥100 Gy whereas genotype Macia was reduced by doses ≥400 Gy. Seed viability was significantly decreased by doses ≥200 and 500 Gy for genotype Macia and Red sorghum, respectively. In general, the result showed that genotype Red sorghum demonstrated stronger tolerance to the tested gamma radiation doses, suggesting that high doses are needed to induce mutagenesis.

## 3. Discussion

### 3.1. Effect of Single Doses of Gamma Radiation

The present study found highly significant interactions between genotypes and gamma radiation doses on assessed traits, viz., germination percentage, seedling emergence percentage, seedling survival percentage, number of panicles per m^2^, number of productive panicles per m^2^, panicle length and seed viability (Table 1 and Table 3). The optimum dose rates calculated on the basis of seedling emergence and survival rates for genotypes Red sorghum, Parbhani Moti, Macia, ICSV 15013 and Parbhani Shakti ranged between 392 and 419 Gy, 311 and 354 Gy, 256 and 355 Gy, 273 and 304 Gy, and 266 and 297 Gy, respectively (Figure 1 and Figure 4). Moreover, results showed that GR50 for shoot length ranged between 436 and 686 Gy, respectively (Figure 1 and Figure 4). However, these dose rates will mean lower seedling emergence, poor seedling survival rate and weak plants. High dose rates produce low plant population and poor plant growth in the M_2_ and M_3_ generations, in turn affecting the identification and selection of useful mutants (Table 2 and Table 4). Golubinova and Gecheff [35] applied gamma radiation dose rates ranging between 100 to 400 Gy on Sudan grass and found that the LD_50_ values estimated on the basis of seedling survival rate were more suitable for large scale mutagenesis than using data on germination and sterility. The authors stressed the importance of using the optimum dose rate which obtains good and healthy plant populations to be grown to maturity. Mudibu et al. [34] suggested that a high mutagen dose rate leads to an increased proportion of changes such as chromosomal aberrations, lethality, injury, and sterility. The present study found that low gamma radiation dose rates (≤200 Gy) showed no significant effect on plant growth except for the germination rate and the number of panicles per m^2^ (Table 4). Meanwhile, a high dose reduced seedling emergence and survival rates, the number of panicles and the number of productive panicles per m^2^, plant height and seed viability below the target of 50% (Table 4). Similar findings were reported in chili pepper [36] and African nightshade [37]. The differences observed in biological traits suggest that the identified optimum doses of single gamma radiation may produce useful mutants, as reported by Tadele [38] and Manova and Gruszka [39].

### 3.2. Effect of Single Doses of EMS

Highly significant interactions were found between genotypes and EMS concentration on seedling emergence and survival rates (Table 1). The LD_50_s calculated on the basis of seedling emergence and survival rate for genotypes Parbhani Shakti, ICSV 15013, Parbhani Moti and Macia was between 0.41% and 0.60%, 0.48% and 0.58%, 0.46% and 0.51%, and 0.36% and 0.45%, respectively (Figure 2). The GR_50_s for shoot length for genotypes ICSV 15013, Parbhani Shakti, Parbhani Moti, and Macia were 1.35%, 0.90%, 0.87% and 0.67% EMS, respectively (Figure 2). GR_50_ may be associated with undesirable effects on growth promoters and various chromosomal aberrations which affect emergence, seedling survival rate and seed set [40]. Therefore, the GR_50_s determined were not suitable for large scale mutagenesis. Mani [17] applied an EMS concentration of 0.3% for 8 h in three sorghum varieties and found germination rates between 63% and 80% and a survival rate between 66% and 82%. This result suggested the need to increase the concentrations to obtain LD_50_s for each genotype. In the present study, the mean seedling emergence, survival rates and seedling height had significant reductions with increased EMS concentrations (Table 2). This result corroborates the findings in rice [41], soybean [34] and wheat [42] which reported that seedling emergence, survival and growth were influenced by genetic variations and EMS concentrations.

### 3.3. Effect of Combined Doses of Gamma Radiation and EMS

Significant interactions were found between genotypes and dose rates for the combined use of gamma radiation followed by EMS concentrations on seedling emergence and survival rates (Table 1). However, results were unable to determine the optimum dose rates for combined applications of gamma radiation and EMS for large scale mutagenesis on the test genotypes. The test dose rates were high and produced undesirable effects, including reduction of seedling emergence and survival rates. The mean of all traits assessed was drastically decreased with the test doses (Table 2). Results showed that, with a combined treatment of 300 Gy followed by 0.1% EMS concentration, seedling emergence percentages of genotypes Parbhani Moti, Macia, Parbhani Shakti and ICSV 15013 were 27.6%, 19.7%, 16.8% and 14.3%, respectively (Figure 3). At the same dose rate, the seedling survival rates of genotypes Parbhani Shakti, Parbhani Moti, Macia, and ICSV 15013 were 16.2%, 15.9%, 3.8% and 3.5%, respectively. These results can be explained by the optimum dose rates obtained in a single use of gamma radiation and EMS on the test genotypes shown above. This result suggests the need to apply lower gamma radiation doses and EMS concentrations to obtain desired level of seedling emergence and survival rates and in turn to promote healthy plant growth for effective selection.

## 4. Materials and Methods

### 4.1. Study Site and Plant Material

The study involved two concurrent experiments carried out under field tests at the International Crops Research Institute for the Semi-Arid Tropics (ICRISAT), Patancheru, India, (17°30′32.9″ N 78°16′50.8″ E) and Mannheim Crop Research Station, Tsumeb, Namibia (19°10′06.7″ S 17°45′54.1″ E). The study used seeds of five sorghum genotypes, viz., Parbhani Moti (SPV 1411), Parbhani Shakti (ICSR 14001), ICSV 15013, Macia (SDS 3220) and Red sorghum. The genotypes were selected for their unique traits; e.g., Macia is one of the popular cultivars in the Southern African region. It is a high-yield semi-dwarf and early-maturity cultivar [43]. Meanwhile, the genotype Red sorghum is a late-maturity cultivar widely cultivated in Namibia [44], and the genotypes Parbhani Shakti, Parbhani Moti and ICSV15013 have unique traits including high contents of grain Fe and Zn and are currently being tested for wider adoption and drought tolerance.

### 4.2. Seed Treatment, Planting and Experimental Design

#### 4.2.1. Experiment I

This study examined the effect of a single or combined use of gamma radiation and EMS. Four sorghum genotypes (‘Parbhani Moti’, ‘Parbhani Shakti’, ‘ICSV 15013′, and ‘Macia’) were subjected to mutagenesis using five doses of gamma radiation (0, 300, 400, 500 and 600 Gy), three EMS doses (0, 0.5 and 1.0%) and subsequent use of gamma radiation followed by EMS (0 and 300 Gy and 0.1% EMS; 400 Gy and 0.05% EMS). Batches of 1000 dry seeds per treatment were irradiated at Bhabha Atomic Research Centre, Nuclear Agriculture and Biotechnology, Mumbai, India. The minimum gamma radiation dose (300 Gy) was selected following favorable responses as previously reported by Bretaudeau [23], Human et al. [45] and Mizuno et al. [24]. EMS dose rates were selected following the germination rate and seedling responses reported by Jiao et al. [46], Mani [17] and Xin et al. [26]. Prior to EMS treatment, 100 seeds per treatment were placed in mesh bags replicated five times for pre-soaking under running tap water for six hours at room temperatures, according to the procedure described by Mba et al. [47] and Ndou et al. [48]. Seeds were removed to dry off excess water and placed in respective prepared EMS solutions for five hours. The seed was rinsed under running tap water for three hours. Treated seeds were sown in 10-m-long single-row plots using a randomized complete block design (RCBD), with five replications in the experimental field of ICRISAT, Patancheru, India. Inter-row spacing of 75 cm and intra-row spacing of 10 cm was used for planting. To ensure adequate soil moisture, supplemental irrigation was provided. Compound synthetic fertilizer as a source of nitrogen (N), phosphorus (P) and potassium (K) was used at the rate of 30 kg N ha^−1^, 45 kg P_2_O_5_ ha^−1^ and 30 kg K_2_O ha^−1^, in that order. The fertilizer was homogeneously broadcasted to the plots before planting.

#### 4.2.2. Experiment II

This experiment examined the effect of gamma radiation from germination to maturity. The experiment involved two genotypes (‘Macia’ and ‘Red sorghum’), which were treated with seven doses of gamma radiation (0, 100, 200, 300, 400, 500 and 600 Gy). The doses of 100 and 200 Gy were included following the result of Experiment I. Batches of 1000 dry seeds per treatment were irradiated at the Joint FAO/IAEA laboratories in Seibersdorf, Austria. Gamma-irradiated seeds were sown in 10-m-long single-row plots laid in an RCBD with three replications in the field at Mannheim Crop Research Station, Namibia. All standard agronomic practices were followed as described above. Urea (46% N) was applied at the rate of 30 kg N ha^−1^ after a 50% emergence.

### 4.3. Data Collection

The following traits were assessed following the methods described by FAO/IAEA [49], Horn and Shimelis [50] and Mba et al. [10]. Germination percentage was recorded by counting sprouted seeds per treatment after 3 days using the moist paper towel germination method under room temperature as described by Sako et al. [51]. Emergence percentage was recorded by counting the number of seedlings raised above ground from sowing to seven days after sowing. Seedling survival percentage was recorded by counting the number of surviving seedlings at 21 days after 50% emergence. The shoot length was recorded from measuring five randomly selected plants per plot from the base to the tip of the top leaf, and the average was expressed in cm 21 days after 50% emergence. The total number of panicles and productive panicles per plot was counted at harvest and expressed in the number of panicles m^−2^. Panicle length was recorded from five randomly selected panicles per plot by measuring from the base to the tip of panicles and expressed in cm. The plant height was measured from five randomly selected plants from the base of the plant to the tip of the panicle and expressed in cm. Seed viability was recorded using a germination test as described above.

### 4.4. Data Analysis

Data on germination percentage, emergence percentage, seedling survival percentage, shoot length, number of panicles m^−2^, number of productive panicles m^−2^, panicle length, plant height and seed viability were subjected to analysis of variance (ANOVA) using Genstat (18th edition) statistical software [52]. Mean comparisons were conducted using Fisher’s least significant difference procedure when significant differences were detected in the ANOVA. The mean lethal dose (LD_50_) and mean growth reduction (GR_50_) were estimated through the simple linear regression model by fitting the straight-line equation y = mx + c; where y is the response variable, x is the independent variable, m and c represent the slope and constant, respectively.

## 5. Conclusions

This study determined doses of gamma radiation and EMS singly or in combination in sorghum based on early growth parameters. Combined doses of gamma radiation and EMS produced poor seedling emergence and seedling survival rate below the target of LD_50_. Therefore, combined doses were not recommended for large scale mutation induction in sorghum. The optimum gamma radiation dose rates aiming at LD_50_ for seedling emergence and survival rate for genotypes Red sorghum, Parbhani Moti, Macia, ICSV 15013 and Parbhani Shakti ranged between 392 and 419 Gy, 311 and 354 Gy, 256 and 355 Gy, 273 and 304 Gy, and 266 and 297 Gy, respectively. The optimum EMS dose rates aiming at LD50 for seedling emergence and survival rate for genotypes Parbhani Shakti, ICSV 15013, Parbhani Moti and Macia were between 0.41% and 0.60%, 0.48% and 0.58%, 0.46% and 0.51%, and 0.36% and 0.45%, respectively. These dose rates may induce genetic variation in the tested sorghum genotypes for genetic enhancement in sorghum breeding programs.

## Figures and Tables

**Figure 1 plants-09-00827-f001:**
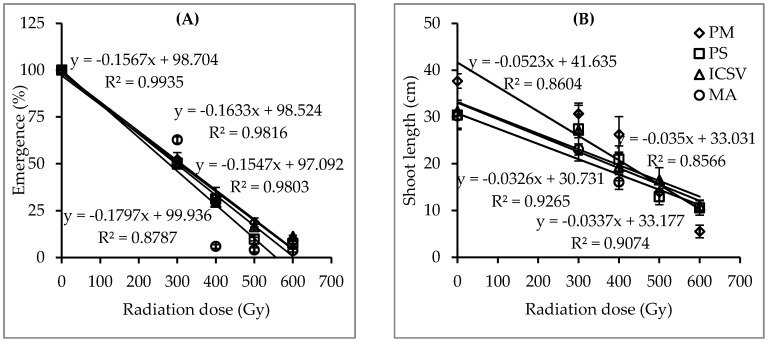
Emergence percentage (**A**) and shoot length (**B**) and fitted straight lines of four sorghum genotypes treated with five gamma radiation doses in experiment I (bars represent standard error). Note: PM = Parbhani Moti, PS = Parbhani Shakti, ICSV = ICSV 15013 and MA = Macia

**Figure 2 plants-09-00827-f002:**
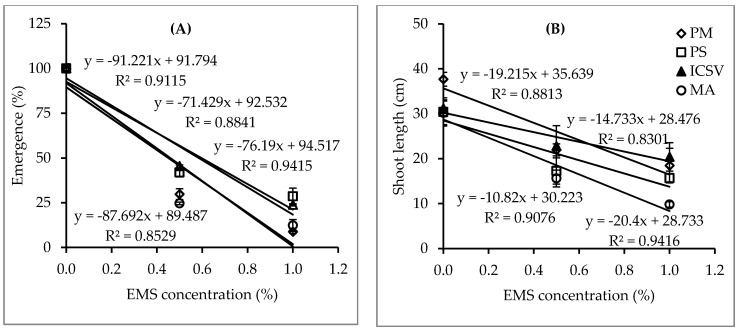
Emergence percentage (**A**) and shoot length (**B**) and fitted straight lines of four sorghum genotypes treated with three EMS concentrations in experiment I (bars represent standard error). Note: PM = Parbhani Moti, PS = Parbhani Shakti, ICSV = ICSV 15013 and MA = Macia

**Figure 3 plants-09-00827-f003:**
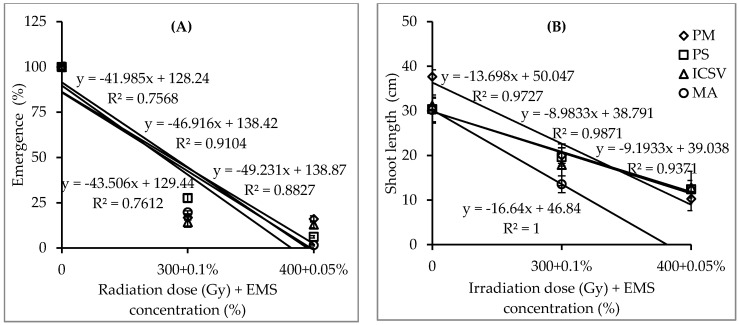
Emergence percentage (**A**) and shoot length (**B**) and fitted straight lines of four sorghum genotypes treated with three sequential combined treatments of gamma followed by EMS in experiment I (bars represent standard error). Note: PM = Parbhani Moti, PS = Parbhani Shakti, ICSV = ICSV 15013 and MA = Macia

**Figure 4 plants-09-00827-f004:**
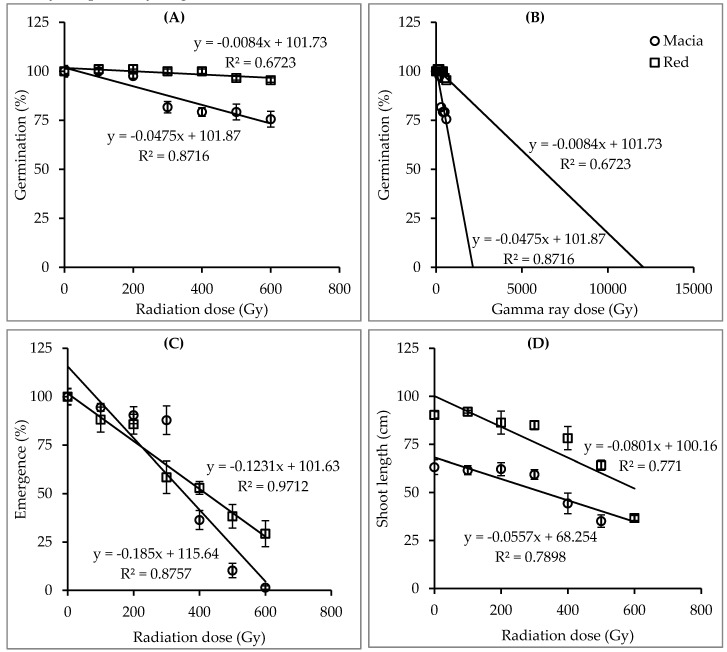
Actual (**A**) and extrapolated (**B**) germination percentage, seedling emergence percentage (**C**), and shoot length (**D**), fitted with straight lines to estimate the LD_50_s of two sorghum genotypes treated with seven gamma radiation doses in experiment II (bars represent standard error). Note: PM = Parbhani Moti, PS = Parbhani Shakti, ICSV = ICSV 15013 and MA = Macia

**Table 1 plants-09-00827-t001:** Mean-square values and significance tests for emergence percentage (%E), seedling survival percentage (%SS) and shoot length (SLT) of four sorghum genotypes evaluated with variable doses of gamma radiation, EMS concentrations and combined treatments in Experiment I.

Mutagenic Treatments	Source of Variation	df	%E		%SS		SLT (cm)	
Gamma radiation	Replication	4	94.6		57.7		30.0	
	Genotype (G)	3	126.7	***	268.2	***	110.5	***
	Dose (D)	4	9415.7	***	6512.8	***	1460.6	***
	Genotype x dose	12	202.7	***	326.9	***	40.8	
	Error	76	16.4		22.8		35.3	
EMS	Replication	4	62.9		62.4		38.1	
	Genotype (G)	3	339	***	275.7	***	173.2	***
	Dose (D)	2	11,831.3	***	9367.7	***	1391.4	***
	Genotype x dose	6	148.8	***	460.2	***	26.3	
	Error	44	22.8		23.5		35.5	
Gamma radiation followed by EMS	Replication	4	49.7		53.5		45.7	
	Genotype (G)	3	185.4	***	431.7	***	58.2	
	Dose (D)	2	15,731.7	***	11,334.6	***	1805.3	***
	Genotype x dose	6	168.9	***	379.4	***	30.2	
	Error	44	11.2		19.3		25	

*** denotes significant differences at *p* ≤ 0.01; df = degree of freedom.

**Table 2 plants-09-00827-t002:** Means of emergence percentage (%E), seedling survival percentage (%SS) and shoot length (SLT) among four sorghum genotypes tested using four gamma radiation, two EMS and two subsequent treatments of gamma and EMS in experiment I.

Mutagen	Dose	Variety	%E		%SS		SLT (cm)	
Gamma radiation	300 Gy	Parbhani Moti	27.40	±3.63	22.20	±3.76	30.70	±2.26
		Parbhani Shakti	31.00	±1.76	24.20	±1.69	27.42	±3.18
		ICSV 15013	23.00	±1.79	14.00	±2.17	27.16	±5.31
		Macia	40.80	±1.66	32.20	±1.83	23.06	±2.48
	400 Gy	Parbhani Moti	16.80	±5.24	14.20	±4.65	26.26	±3.79
		Parbhani Shakti	19.20	±0.58	10.60	±1.60	20.92	±2.88
		ICSV 15013	13.60	±1.47	8.60	±1.44	19.34	±2.99
		Macia	3.80	±1.02	0.80	±0.49	16.08	±1.60
	500 Gy	Parbhani Moti	9.80	±2.35	2.00	±0.32	13.94	±1.92
		Parbhani Shakti	6.00	±0.95	1.60	±0.40	12.93	±1.69
		ICSV 15013	7.60	±1.33	1.80	±0.86	16.50	±2.64
		Macia	2.60	±1.12	0.00	±0.00	-	
	600 Gy	Parbhani Moti	4.40	±0.81	1.60	±0.24	5.50	±1.35
		Parbhani Shakti	4.60	±1.94	1.20	±0.49	10.53	±1.53
		ICSV 15013	5.20	±0.58	1.80	±0.58	11.00	±1.22
		Macia	2.40	±0.81	0.00	±0.00	-	
EMS	0.50%	Parbhani Moti	15.60	±3.01	11.20	±2.03	21.96	±1.33
		Parbhani Shakti	25.80	±2.42	7.80	±2.18	17.26	±2.94
		ICSV 15013	21.00	±0.71	10.60	±1.17	22.82	±4.51
		Macia	16.00	±0.71	4.20	±0.49	15.60	±1.88
	1%	Parbhani Moti	4.60	±0.87	3.00	±0.32	18.46	±3.86
		Parbhani Shakti	17.60	±4.62	6.40	±2.18	15.67	±1.05
		ICSV 15013	11.00	±0.63	4.80	±1.07	20.40	±3.15
		Macia	8.00	±3.35	2.40	±1.69	9.80	±0.60
Gamma radiation followed by EMS	300 Gy + 0.1% EMS	Parbhani Moti	8.80	±1.36	4.40	±0.68	20.00	±1.74
		Parbhani Shakti	17.00	±2.00	9.00	±2.49	19.64	±2.04
		ICSV 15013	6.60	±2.82	1.20	±0.73	17.90	±4.67
		Macia	12.80	±0.86	2.20	±0.37	13.56	±1.88
	400 Gy + 0.05% EMS	Parbhani Moti	8.40	±1.72	1.40	±0.51	10.28	±2.64
		Parbhani Shakti	3.80	±0.66	0.60	±0.40	12.43	±1.97
		ICSV 15013	6.00	±1.82	1.00	±0.77	12.83	±3.61
		Macia	1.00	±0.45	0.00	±0.00	-	
Control	0	Parbhani Moti	52.40	±1.12	27.60	±4.80	37.68	±1.54
		Parbhani Shakti	61.60	±1.36	55.60	±2.94	30.40	±3.15
		ICSV 15013	46.20	±2.58	34.00	±2.76	31.22	±1.85
		Macia	65.00	±1.82	57.20	±2.35	30.20	±2.63
Grand Mean			17.4		10.6		19.4	
LSD (5%)			5.3		4.9		2.7	
CV (%)			24.3		37.3		29.9	
R^2^ (%)			83.6		77.5		53.8	

± denotes standard error.

**Table 3 plants-09-00827-t003:** Mean-square values and significance tests for germination percentage (%G), seedling emergence percentage (%E), seedling survival percentage (%SS), shoot length (SLT), number of panicles per m^2^ (NP), number of productive panicles per m^2^ (NPP), panicle length (PLT), plant height (PHT) and seed viability (%SV) of two sorghum genotypes evaluated with variable doses of gamma radiation in experiment II.

**Source of Variation**	**df**	**%G**		**%E**		**%SS**		**SLT (cm)**		**NP**	
Replication	2	13.0		3.3		4.5		66.1		1.2	
Genotype (G)	1	3498.7	***	92.7	***	398.7	***	7238.2	***	151.2	***
Dose (D)	6	205.8	***	1515.2	***	1174.9	***	1494	***	56.1	***
G x D	6	109.2	***	132.3	***	53	***	18.6		12.3	***
Error	26	13.49		3.3		3.8		36		0.9	
**Source of variation**	**df**	**NPP**		**PLT (cm)**		**PHT (cm)**		**%SV**			
Replication	2	0.6		6.8		77.7		25.9			
Genotype (G)	1	105.4	***	3.3		25834.6	***	5208	***		
Dose (D)	6	50.4	***	10.5	***	1252.1	***	1187.3	***		
G x D	6	4.6	***	13	***	78.1		456	***		
Error	26	0.8		2.4		42		76.3			

*** denotes significant differences at *p* ≤ 0.01; df = degree of freedom.

**Table 4 plants-09-00827-t004:** Mean germination percentage (%G), emergence percentage (%E), seedling survival percentage (%SS), shoot length (SLT), plant height (PHT), number of panicles per m^2^ (NP), number of productive panicles per m^2^ (NPP), panicle length (PLT) and seed viability (%SV) of sorghum involving six gamma irradiations using three replications in experiment II.

Genotype	Dose (Gy)	%G		%E		%SS		SLT (cm)		PHT (cm)		NP		NPP		PLT (cm)		%SV	
Macia	0	91.1	±2.9	46.0	±4.4	41.5	±5.2	63.2	±3.8	137.8	±3.2	8.6	±4.4	5.2	±2.6	17.6	±1.2	80.0	±2.9
	100	91.1	±1.1	43.5	±2.0	38.9	±5.2	61.6	±2.3	133.4	±4.1	8.5	±4.4	7.1	±2.1	16.7	±0.1	71.7	±1.7
	200	88.9	±1.1	41.6	±4.4	27.7	±4.4	62.2	±3.4	132.1	±3.3	8.2	±2.4	6.8	±3.3	17.2	±2.2	70.0	±2.9
	300	74.4	±2.9	40.4	±7.4	16.1	±7.3	59.4	±2.5	129.4	±3.8	7.6	±5.5	5.3	±4.9	17.7	±1.7	58.3	±6.0
	400	72.2	±2.2	16.7	±4.9	12.3	±2.8	44.4	±5.3	112.5	±4.4	3.3	±1.7	1.2	±0.3	20.1	±0.5	51.7	±7.3
	500	72.2	±4.0	4.7	±3.8	2.2	±1.5	35.1	±3.2	96.2	±4.8	1.0	±1.7	0.2	±0.9	23.3	±0.5	14.0	±8.3
	600	68.9	±4.0	0.7	±0.9	0.0	±0.0	-		-		-		-		-		-	
Red sorghum	0	98.9	±1.1	47.3	±4.1	42.3	±8.5	90.3	±1.7	197.7	±2.8	8.5	±6.0	7.0	±14.8	20.2	±0.1	88.3	±4.4
	100	100.0	±0.0	41.7	±6.5	35.8	±7.9	92.0	±0.8	184.2	±2.7	14.8	±4.3	12.1	±31.0	16.8	±0.6	88.3	±3.3
	200	100.0	±0.0	40.6	±5.1	31.9	±5.6	86.3	±6.0	176.6	±1.7	10.0	±3.2	8.4	±19.7	17.6	±0.8	85.0	±2.9
	300	98.9	±1.1	27.7	±8.4	24.0	±7.2	85.0	±2.4	176.4	±3.5	8.5	±5.8	6.5	±16.8	19.4	±0.4	80.0	±2.9
	400	98.9	±1.1	25.1	±3.3	22.1	±3.3	78.2	±6.0	169.6	±6.2	8.6	±6.1	6.3	±16.6	16.7	±0.7	76.7	±1.7
	500	95.6	±1.1	18.1	±6.1	16.6	±7.4	64.2	±2.6	158.4	±5.1	8.1	±1.0	4.8	±11.9	18.4	±0.8	71.7	±8.8
	600	94.4	±1.1	13.9	±6.8	9.2	±5.3	36.8	±1.3	153.7	±2.0	5.2	±4.3	2.8	±3.0	19.1	±0.1	70.0	±2.9
Grand Mean		89.0		29.1		22.9		66.1		150.6		7.8		5.7		18.5		69.7	
LSD (5%)		6.2		3.0		3.3		4.0		4.3		0.6		0.6		1.0		5.8	
CV (%)		4.1		6.2		8.5		9.1		4.3		12.5		16.1		8.4		12.5	
R^2^ (%)		87.8		88.9		96.3		77.6		93.7		62.9		62.5		48.4		83.0	

± denotes standard error.

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
