# Peer review of "The Effect of Single and Combined Use of Gamma Radiation and Ethylmethane Sulfonate on Early Growth Parameters in Sorghum"

_plants, 2020, doi:10.3390/plants9070827_

Round 1

Reviewer 1 Report

The introduction should have relevant references in breeding needs and breeding achievement in sorghum. With the information, the specific needs should be addressed why the mutation breeding method elaboration is needed here in this manuscript.

Plagiarism self checking is appreciated by using any of softwares.

The novelty is questionable as mutation breeding itself is an old approach which have been tested for decades for a plant biology journal. However, for the area of Plant Breeding this may add some references for alternative approaches.

Experimental design is typical factorial and one more integrative multivariate assessment could make a simpler interpretation.

The reference section is rather mess: no volume and no page of many journals in many references Also no constant style rule fit with Plants. This must be intensively edited, otherwise, the credibility of the information is decreased.

The use of word, "ideal" would have comparative measurement, and the use of some words should be reconsidered such for "appropriate / optimum". Likewise, some expression of words shall be carefully edited.

The combination of radiation and chemical mutation would be variable depending on the plant material conditions and genotypes, and pedigree information also may help, how the result could be applied to other genotypes or populations. 

Author Response

27th May 2020

Ms. Lavinia Bianca Balea
Assistant Editor

Plants,

MDPI OPEN ACCESS PUBLISHING ROMANIA SRL

Dear Ms. Balea

REF: Submission of a revised manuscript ID: plants-812291

Thank you for the reviewers’ and associate editors’ valuable comments on our Manuscript ID: plants-812291 entitled “The Effect of Single and Combined Use of Gamma Radiation and Ethylmethane Sulfonate in Sorghum Mutagenesis”. We have carefully revised all suggested comments, which significantly improved the earlier version of the manuscript.

Below are point-by-point explanations on suggested comments, which are indicated in track change mode in the current submission.

With kind regards

Maliata Athon Wanga

Comment: The introduction should have relevant references in breeding needs and breeding achievement in sorghum. With the information, the specific needs should be addressed why the mutation breeding method elaboration is needed here in this manuscript.

  • Response: As advised, references for various breeding objectives have been included on Page 2, line 60-76.

Comment: Plagiarism self-checking is appreciated by using any of softwares.

  • Response: Similarity checked performed using an online Plagiarism Checker.

Comment: The reference section is rather mess: no volume and no page of many journals in many references Also no constant style rule fit with Plants. This must be intensively edited; otherwise, the credibility of the information is decreased.

  • Response: As advised, we have edited the reference section.

Comment: The use of word, "ideal" would have comparative measurement, and the use of some words should be reconsidered such for "appropriate / optimum". Likewise, some expression of words shall be carefully edited.

  • Response: The word “optimum” is used.

Reviewer 2 Report

In this study the effect of a single and combined gamma radiation and ethyl methane sulfonate (EMS) in five sorghum genotypes was determined. These treatments influenced the germination percentage, the seedling emergency percentage, survival rate of seedling, number of panicles, number of production cobs, cob length and seed viability. This is an agronomic study and there is nothing about mutagenesis. We cannot speak of mutagenesis if no genomic study has been conducted. I would talk about the effects of these treatments on the growth of sorghum plants. Mutagenesis experiments are completely different. The works that the authors cite (for example Mizuno et al. 2013, Xin et al. 2008, Jiao et al. 2018 etc.) report the effects of mutagenic agents on the genome.I would ask the authors to remodel the work focusing only on the effects of mutagenic substances on morphological and physiological changes.

Author Response

27th May 2020

Ms. Lavinia Bianca Balea
Assistant Editor

Plants,

MDPI OPEN ACCESS PUBLISHING ROMANIA SRL

Dear Ms. Balea

REF: Submission of a revised manuscript ID: plants-812291

Thank you for the reviewers’ and associate editors’ valuable comments on our Manuscript ID: plants-812291 entitled “The Effect of Single and Combined Use of Gamma Radiation and Ethylmethane Sulfonate in Sorghum Mutagenesis”. We have carefully revised all suggested comments, which significantly improved the earlier version of the manuscript.

Below are point-by-point explanations on suggested comments, which are indicated in track change mode in the current submission.

With kind regards

Maliata Athon Wanga

Comment: This is an agronomic study and there is nothing about mutagenesis. We cannot speak of mutagenesis if no genomic study has been conducted. I would talk about the effects of these treatments on the growth of sorghum plants. I would ask the authors to remodel the work focusing only on the effects of mutagenic substances on morphological and physiological changes.

  • Response: We have revised the title as follows: “The effect of single and combined use of gamma radiation and ethylmethane sulfonate on early growth parameters in sorghum”. The focus of the study was to compare the effectiveness of the use of the two techniques in inducing genetic variation in sorghum using early growth parameters.

Reviewer 3 Report

Review of Plants- 812291

The effect of single and combined use of gamma radiation and ethylmethane sulfonate in sorghum mutagenesis

Maliata Athon Wanga, Hussein Shimelis, Lydia N. Horn, and Fatma Sarsu

The authors treated seeds of several different sorghum genotypes with various doses of gamma radiation and/or ethylmethane sulfonate (EMS) on the germination rates and several aspects of their subsequent performance including emergence, survivorship, plant height and number of productive panicles. The goal was to identify dosages which resulted in mean lethal doses of 50% (LD50) and mean growth reduction (GR50) of 50%, as previous studies had shown that doses with these outcomes were more likely to render useful mutagenic events. The authors found that the various sorghum genotypes differed in the dosages required to achieve these outcomes, where “Red sorghum” required the highest dosage while “Macia” required the lowest dosage.

This study provides useful information worth sharing with the plant community after correcting the problems noted below. Perhaps the most important is that the tables and figures should include indicators of variability. That is, the averages listed in the tables should also indicate the standard deviation or the standard error, and the figures should show error bars, and their captions should indicate whether these represent the standard deviation or the standard error as well as the sample size (i.e. n = ??). In addition, the figure captions should quickly summarize the “Materials and Methods” and explain the symbols.

The figures and tables should be integrated with the text rather than presented as separate items below the text. For example, Tables 1 and 2 and figure 1 should be integrated into section 2.1.1 rather than listed below in Section 2.2.

How did the authors specify the ideal EMS dosages with such precision (e.g. line 29) when they only tested concentrations of 0,5 and 1%?

I am puzzled that the authors didn’t include grain yield per plant, since they measured the numbers of panicles, productive panicles, and seed survivorship. This seems like another useful measure of how viable the plants surviving these treatments are.

It would also be useful to note in the introduction and in the discussion that gamma irradiation and EMS produce different kinds of mutations. Gamma irradiation breaks the DNA resulting in deletions, translocations and chromosomal rearrangements, whereas EMS modifies bases and therefore usually causes missense or nonsense mutations. The goals of the breeding program might favor one mutagen over the other even if gamma irradiation appears to yield a more useful range of LD50 and GR50 values.

This would have been a much stronger paper if it had included some measure of mutation frequency in the treated populations. This would be more challenging, but they could have measured the frequency of known mutations or the frequency of albinism in the F2 generation after selfing. This would have helped determine whether the goals of LD50 and GR50 are in fact a good target for mutagenesis studies.

Overall, the English is quite good, but there are numerous errors that require correction.

Some specific examples of sentences that require clarification include:

Lines 23-25 are hard to understand and should be rewritten. In addition, the authors should clarify what they mean by the “best dosage.”

Line 41 is hard to understand and should be rewritten.

Line 56 should be “… mutagens such as ethylmethane…”

Line 64 should be “…EMS doses ranging …”

Line 66 should be “…sorghum were produced when the seed was treated with EMS doses ranging …”

Line 71 should be “…The mutant variety …”

Line 73 is grammatically incorrect and should be rewritten.

Lines 77-78 should be “…EMS was used to generate 107 of the mutant plant …”

Lines 80-83 are grammatically incorrect and should be rewritten.

Line 95 is hard to understand and should be rewritten. Do you mean “Effect of single dose of gamma radiation?” Or “Effect of gamma irradiation by itself?”

Lines 106-107 are hard to understand and should be rewritten.

Line 114 is hard to understand and should be rewritten. Do you mean “Effect of a single dose of EMS?” Or “Effect of EMS by itself?”

Line117: since you did not measure mutation rates you can’t say that this result shows that the ability of EMS to induce mutations is responsible for this result. Perhaps EMS harms the treated seeds in other ways, just as ethidium bromide inhibits protein synthesis in addition to causing mutations.

Line 148 “number of panicle” should be “number of panicles” and “number of productive panicle” should be “number of productive panicles”

Line 149 “Table 4 summarizes the mean of traits” should be “Table 4 summarizes the means of traits”

Lines 151-153 are grammatically incorrect and should be rewritten.

Line 155 Please rewrite for clarity

Line 162 “number of panicle” should be “number of panicles”

Line 169 should be “high doses are needed”

Lines 172 and 178 should be “…and significance tests…”

Line 175 should be “Means of…”

Lines 204 and 231: as noted above, please clarify whether they mean that gamma radiation or EMS were used by themselves, or used only once?

Lines 208-209 are grammatically incorrect and should be rewritten.

Lines 214-216 are grammatically incorrect and should be rewritten.

Lines 219-221 are hard to understand and should be rewritten.

Lines 223-227 are hard to understand and should be rewritten (and please correct “assed”)

Lines 233-234 are grammatically incorrect and should be rewritten.

Lines 270-275 are grammatically incorrect and should be rewritten

Lines 331-334 are grammatically incorrect and should be rewritten

Author Response

27th May 2020

Ms. Lavinia Bianca Balea
Assistant Editor

Plants,

MDPI OPEN ACCESS PUBLISHING ROMANIA SRL

Dear Ms. Balea

REF: Submission of a revised manuscript ID: plants-812291

Thank you for the reviewers’ and associate editors’ valuable comments on our Manuscript ID: plants-812291 entitled “The Effect of Single and Combined Use of Gamma Radiation and Ethylmethane Sulfonate in Sorghum Mutagenesis”. We have carefully revised all suggested comments, which significantly improved the earlier version of the manuscript.

Below are point-by-point explanations on suggested comments, which are indicated in track change mode in the current submission.

With kind regards

Maliata Athon Wanga

Comment: Perhaps the most important is that the tables and figures should include indicators of variability. That is, the averages listed in the tables should also indicate the standard deviation or the standard error, and the figures should show error bars, and their captions should indicate whether these represent the standard deviation or the standard error as well as the sample size (i.e. n = ??). In addition, the figure captions should quickly summarize the “Materials and Methods” and explain the symbols.

  • Response: As advised, a standard error has been added to the tables and figures.

Comment: The figures and tables should be integrated with the text rather than presented as separate items below the text. For example, Tables 1 and 2 and figure 1 should be integrated into section 2.1.1 rather than listed below in Section 2.2.

  • Response: We have rearranged the figures and tables according to sections.

Comment: How did the authors specify the ideal EMS dosages with such precision (e.g. line 29) when they only tested concentrations of 0,5 and 1%?

  • Response: We recommended the doses based on the calculated LD50 of emergence and survival rate.

Comment: I am puzzled that the authors didn’t include grain yield per plant, since they measured the numbers of panicles, productive panicles, and seed survivorship. This seems like another useful measure of how viable the plants surviving these treatments are.

  • Response: Thank you for this valuable observation. This study only focused on early growth-related parameters.

Comment: It would also be useful to note in the introduction and in the discussion that gamma irradiation and EMS produce different kinds of mutations. Gamma irradiation breaks the DNA resulting in deletions, translocations and chromosomal rearrangements, whereas EMS modifies bases and therefore usually causes missense or nonsense mutations. The goals of the breeding program might favor one mutagen over the other even if gamma irradiation appears to yield a more useful range of LD50 and GR50 values.

  • Response: Thank you for the comment. It is revised, lines 50 to 53.

Comment: This would have been a much stronger paper if it had included some measure of mutation frequency in the treated populations. This would be more challenging, but they could have measured the frequency of known mutations or the frequency of albinism in the F2 generation after selfing. This would have helped determine whether the goals of LD50 and GR50 are in fact a good target for mutagenesis studies.

  • Response: Thank you. This will be included in future studies.

Comment: Overall, the English is quite good, but there are numerous errors that require correction. Some specific examples of sentences that require clarification include:

Lines 23-25 and 41 are hard to understand and should be rewritten. In addition, the authors should clarify what they mean by the “best dosage.”

  • Response: The sentence has been revised (Line 24-25).

All other suggested comments and edits are incorporated as suggested.

Round 2

Reviewer 2 Report

The paper in present form can be accepted for me.

Best regard

Reviewer 3 Report

The authors have satisfactorily addressed all of the major concerns raised in my first review. There are still many English errors that should be corrected before it is published.